# An optofluidic antenna for enhancing the sensitivity of single-emitter measurements

Luis Morales-Inostroza[1,2,3,6], Julian Folz [4,6], Ralf Kühnemuth[4], Suren Felekyan [4], Franz-Ferdinand Wieser[1,2,3], Claus A. M. Seidel [4] ✉, Stephan Götzinger [1,3,5] & Vahid Sandoghdar [1,3] ✉

Many single-molecule investigations are performed in fluidic environments, for example, to avoid unwanted consequences of contact with surfaces. Diffusion of molecules in this arrangement limits the observation time and the number of collected photons, thus, compromising studies of processes with fast or slow dynamics. Here, we introduce a planar optofluidic antenna (OFA), which enhances the fluorescence signal from molecules by about 5 times per passage, leads to about 7-fold more frequent returns to the observation volume, and significantly lengthens the diffusion time within one passage. We use single-molecule multi-parameter fluorescence detection (sm-MFD), fluorescence correlation spectroscopy (FCS) and Förster resonance energy transfer (FRET) measurements to characterize our OFAs. The antenna advantages are showcased by examining both the slow (ms) and fast (50 $\mu$s) dynamics of DNA four-way (Holliday) junctions with real-time resolution. The FRET trajectories provide evidence for the absence of an intermediate conformational state and introduce an upper bound for its lifetime. The ease of implementation and compatibility with various microscopy modalities make OFAs broadly applicable to a diverse range of studies.

Since its debut in the early 1990s, single-emitter fluorescence detection has found applications in many fields, ranging from biophysics to quantum optics[1–5]. Dynamic and sensitive measurements under fluidic conditions, however, remain challenging because diffusion restricts the observation time and the detected photon counts, hampering the investigation of both slow and fast phenomena. One can immobilize biomolecules on functionalized surfaces, albeit at the risk of influencing the conformational dynamics. To extend the observation time without jeopardizing contact with other objects, various trapping and confinement schemes have been explored[6–9]. Furthermore, optical microresonators and plasmonic nano-antennas have been employed to enhance the detection rate although these approaches are usually only effective within spectral resonances and require nanometric

positioning of the fluorophore close to surfaces[10,11]. As a result, quantitative studies of many biophysical and biochemical phenomena involving conformational changes of proteins remain difficult[2].

In the past decade, we have proposed and demonstrated a versatile planar antenna that provides collection efficiencies as high as 99% over a very broad spectral domain[12–15]. In this work, we demonstrate how this concept can be implemented in fluidic arrangements for enhancing the collected number of photons from a single nano-object. We show that the architecture of an optofluidic antenna (OFA) combines three separate effects. The optical design adopted from a planar dielectric antenna[12] enhances the fluorescence signal detected from an emitter by about 4.7 times per passage. Moreover, the integrated fluid-gas interface in the OFA can substantially lengthen the

[1]Max Planck Institute for the Science of Light, 91058 Erlangen, Germany. [2]Max-Planck-Zentrum für Physik und Medizin, 91058 Erlangen, Germany. [3]Department of Physics, Friedrich-Alexander-Universität Erlangen-Nürnberg, 91058 Erlangen, Germany. [4]Chair for Molecular Physical Chemistry, Heinrich Heine University Düsseldorf, 40225 Düsseldorf, Germany. [5]Erlangen Graduate School in Advanced Optical Technologies (SAOT), Friedrich-Alexander-Universität Erlangen-Nürnberg, D-91052 Erlangen, Germany. [6]These authors contributed equally: Luis Morales-Inostroza, Julian Folz. ✉ e-mail: cseidel@hhu.de; vahid.sandoghdar@mpl.mpg.de

diffusion time within one passage (7.5 fold in this article). Furthermore, the quasi-two-dimensional geometry of an OFA leads to about 7 times more frequent returns to the observation volume. We demonstrate the virtues of this new technology by performing fluorescence correlation spectroscopy (FCS) and single-molecule (sm) Förster resonance energy transfer (FRET) measurements on dye molecules and DNA-4-way junctions (DNA-4WJ). The ease of implementation and operation of OFAs heralds them as a convenient platform for achieving more powerful sm-fluorescence measurements.

## Results

### Theory and design of an optofluidic antenna

The underlying physics of a dielectric planar antenna is based on reshaping the radiation pattern of an emitter by means of a stratified dielectric structure[12]. This idea has some similarity to the redistribution of dipolar emission in the evanescent field of an interface[10]. However, those approaches can maximally collect up to 40% of the radiation from a randomly oriented dipole and require the emitter to be within $\lambda/2\pi$ of the interface[13]. In the simplest arrangement of a dielectric planar antenna, an emitter is placed within a medium with refractive index $n_2$, sandwiched between two media of refractive indices $n_1$ and $n_3$ such that $n_1 > n_2 > n_3$. In an intuitive picture, the emitter radiation is channeled into the middle layer ($n_2$) which then leaks to the substrate with the higher index. The thickness of medium 2 is correspondingly chosen to be on the scale of the emission wavelength ($\lambda$), but the performance of the device remains tolerant to deviations in this quantity and to the exact position of the emitter in this layer. Another noteworthy feature of this antenna is a large spectral bandwidth spanning over more than three hundred nanometers in wavelength[13].

Experimental demonstrations of planar antennas have previously focused on solid-state emitters and applications in quantum optics[12,14,15]. In this work, we introduce the OFA for applications in biophysical and biochemical studies, where sensitive fluorescence measurements at a high photon flux are needed to investigate the dynamics of molecular machines manifested by conformational exchange. A technical challenge in the realization of an OFA is to maintain the thickness of the fluidic medium 2 in the range of one micrometer or less. While nanofluidic chips can fulfill this condition between two solid media[7], realization of a thin aqueous medium bordering a material of lower refractive index is nontrivial. To address this issue, we have devised a setup for controlled and local confinement of a liquid between a solid substrate and a gaseous boundary.

The schematics of our OFA is shown in Fig. 1a, and more details are provided in Supplementary Note 1. A heat-pulled tapered micropipette

with an inner diameter in the range of 10 μm is dipped in an aqueous buffer and brought close to the underlying substrate. For the experiments reported in this work, we used both commercially available micropipettes and those heat pulled in our laboratory. A piezo actuator is used to adjust and tune the height of the resulting fluidic channel. By chemically modifying the inner wall of the micropipette (see Methods), we prevent the liquid from wetting it from below, thus accommodating a gaseous upper medium within the micropipette. Additionally, a syringe connected to the non-tapered end of the pipette is used to adjust the gas pressure inside it and hence control the shape of the water meniscus at the micropipette opening. We use a standard cover glass as medium 1 in combination with an immersion microscope objective.

The curves in Fig. 1b display the normalized power density as a function of the opening angle of the collection optics, and Fig. 1c shows the fraction of the total number of emitted photons detected up to a certain collection angle. The insets in Fig. 1c sketch the radiation pattern of a dipole averaged over all orientations (i.e., assuming fast molecular rotation) in a homogeneous open solution (blue) and in an OFA (red). We find that an average photon collection efficiency as high as 86% can be achieved for a fast rotating dipole in an OFA, corresponding to nearly 2.2 fold more detected photons than in open solution. In addition to directing the emitted photons toward the detector[12,14,15], the quasi-two-dimensional geometry of OFA also brings about other significant merits, which we discuss below.

### Enhancing the sensitivity of fluorescence measurements: an FCS study

OFAs can be used in a number of applications where higher photon budgets are advantageous. To demonstrate the main features of this methodology, we discuss FCS measurements, where a single particle or molecule diffusing in a liquid through a focused laser beam generates a fluorescence "burst". The area under such a burst indicates the cumulative number of detected fluorescence photons $N_{tot}$ during one passage. A quantitative analysis of the intensity autocorrelation function computed for the detected photons allows one to deduce the number of emitters that contribute to the fluorescence signal (see Supplementary Note 2). In the case, where bursts from multiple molecules overlap, one defines a "molecular brightness Q" to denote the number of detected photons per second and molecule (see Supplementary Note 2, Eq. 3). A pinhole in the detection path restricts the origin of the signal to a quasi-ellipsoidal volume[16,17]. The illumination focus parameter and the pinhole are typically adjusted to obtain an observation volume that is larger than the minimal value of a tight

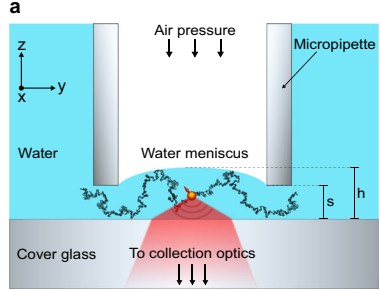

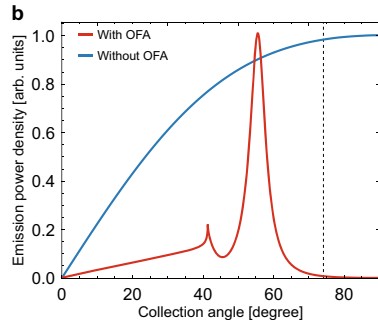

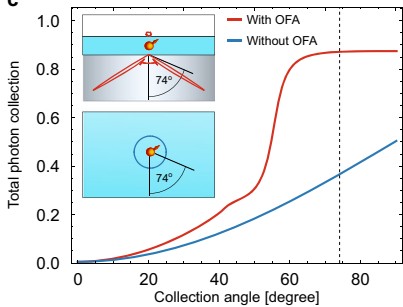

**Fig. 1 | Optofluidic antenna designed for 86% photon collection efficiency from an emitter with fast rotating dipole. a** Sketch of a single molecule diffusing (black curve) through an OFA. A micropipette with inner diameter and wall thickness 14 μm and 3 μm, respectively, forms a thin water layer on a cover glass. The layer thickness in the middle of the resulting meniscus reads $h$ - 500 nm when the distance between the micropipette walls and the cover glass is set to $s = 100$ nm. The indices of refraction are $n_1 = 1.0$ (medium 1, air), $n_2 = 1.33$ (medium 2, water) and $n_3 = 1.517$ (medium 3, glass). **b** Normalized power density plotted as a function of the collection angle. **c** Fraction of the total emitted power integrated up to a given angle averaged for a randomly oriented dipole. Vertical black dashed line in (**b**, **c**) indicate the maximum collection angle of the microscope objective (74°). Blue and red curves represent the situations for an open solution and OFA, respectively. The insets show the respective radiation patterns of a dipole averaged over all orientations in a polar representation for the optofluidic antenna (top panel) and in open solution (bottom panel).

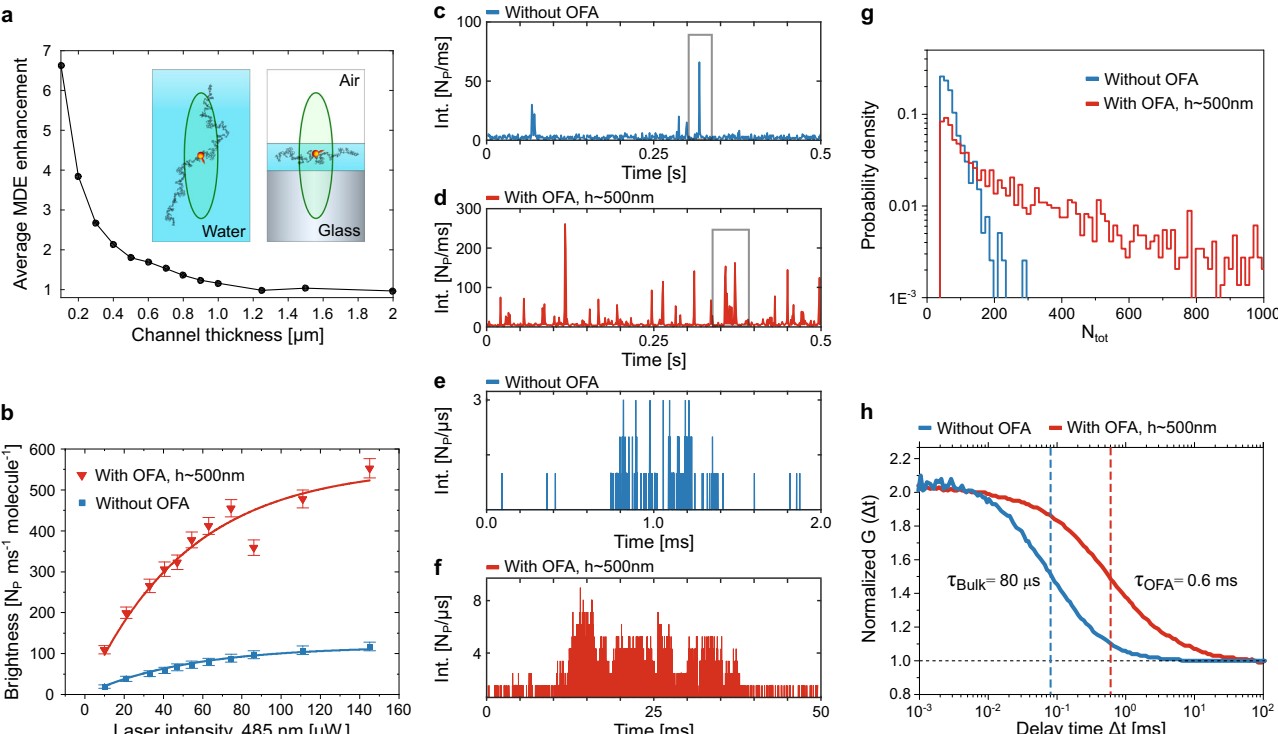

**Fig. 2 | Fluorescence correlation spectroscopy in OFA. a** Relative enhancement of the molecular detection efficiency (MDE) as a function of the water channel thickness $h$. The observation volume is centered at $h/2$. Insets illustrate the situation of the observation volume and the diffusion space for the cases of open solution and OFA. **b** Power dependent brightness in confocal measurements of freely diffusing Rhodamin110 molecules obtained via FCS for the case of an open solution (blue curve and data points) and with the OFA (red curve and data points). Each data point represents a 10 min measurement, and the error bars correspond to the standard deviation of the fluorescence signal (see Supplementary Note 2). The brightness is stated as the ratio of the obtained signal divided by the number of

molecules in a bright state (triplet corrected) within the focus. The model fitted a global value of the saturation power of 64 μW. **c, d** Time trace recorded in a standard FCS arrangement in an open solution (**c**) and in an OFA with $h = 500$ nm (**d**). **e, f** Close-up views of exemplary bursts from (**c, d**). **g** Normalized histogram of the total number of photons ($N_{tot}$) obtained per burst in an open solution (blue bars), and with the OFA (red bars). **h** Typical autocorrelation curves obtained when analytes diffuse in an open solution (blue curve) and inside the OFA (red curve). Measurements were performed using a microscope objective with a numerical aperture $NA = 1.4$.

diffraction-limited focus. FCS detection can be quantified by the molecular detection efficiency function (MDE), defined as the collection efficiency of the optical system multiplied by the excitation efficiency (see Supplementary Note 3).

As depicted in the inset of Fig. 2a, in an OFA the observation volume is considerably reduced in the axial direction, forcing every analyte molecule to pass through the maximum of the detection volume aligned with the pinhole. Consequently, the average molecular brightness is larger in an OFA than in open solution. To gain more insight into this phenomenon, we used analytical calculations to solve for the emission pattern of a dye located at an arbitrary position under the condition that its emitted photons pass through the detection pinhole (see Methods and Supplementary Information). The symbols in Fig. 2a show the resulting MDE enhancement factor as compared to the case of an open liquid. We find an enhancement of about 1.8 times for an observation volume that is axially centered in a channel with $h = 500$ nm. The enhancement factor drops to 1 for water layer thickness larger than $\approx 1$ μm.

To examine this optical confinement effect experimentally, we varied the water channel thickness by adjusting the distance of the micropipette end from the glass substrate. In addition, by varying the excitation power and recording saturation curves, we monitored the effective excitation power. The symbols in Fig. 2b present the outcome for the conventional FCS arrangement in open solution (blue) and an OFA with channel thickness $h = 500$ nm (red). The solid curves (see Supplementary Note 2, Eq. 4) fitted to the experimental data show that the fluorescence signal follows the expected saturation

curve. We verified a 4.7-fold enhancement in brightness independently of the excitation power (see also Supplementary Information). This enhancement results from the combined effects of the increased collection efficiency in our OFA (slightly larger than 2.2) and an effective reduction of the observation volume (see Fig. 2a), which we deduce to be a factor of 4.7:2.2 = 2.1, in good agreement with the outcome of calculations.

In Fig. 2c, d we display examples of fluorescence bursts recorded over a period of 500 ms from Rhodamin 110 (Rh110) dye molecules diffusing in water for the case of an open solution (c) and in an OFA (d). Figure 2e, f portrays close-ups of strong bursts from (c,d). We remark in passing that since each molecule diffuses through a different stochastic path, bursts can vary in duration and in the number of photons, but the histograms in Fig. 2g clearly show that $N_{tot}$ per burst is significantly larger in an OFA[18]. A visual comparison of the data in Fig. 2c, e with those presented in Fig. 2d, f indicates that in the latter case (1) the average brightness is larger (i.e., the number of photons per unit of time is higher), (2) the bursts last longer, and (3) the overall number of recorded bursts is considerably larger than in the case of diffusion in an open solution. While the first effect stems from optical phenomena, the latter two observations have their origin in the microfluidic arrangement of the OFA, as we discuss below.

To scrutinize the duration of fluorescence bursts, in Fig. 2h we display normalized second-order autocorrelation functions $G(\Delta t)$ of the fluorescence signal originating from the observation volume. The observed correlation time $t_c = 80$ μs suggests an average diffusion time of about 0.1 ms for a molecule in open solution (blue curve) if one

models the observation volume as a 3D Gaussian shape with lateral and axial extents of $1\,\mu m$ and $4\,\mu m$, respectively ($1/e^2$ values)[19]. The red curve in Fig. 2h shows that the correlation time and therefore the passage time are substantially stretched (here by 7.5 times) in the OFA arrangement. Monte Carlo simulations indicate that a residence time of $40\,\mu s$ upon each water-air interface visit reproduces our experimental observations.

We attribute the slow diffusion in a thin water layer to the affinity of dye molecules to the water-air interface, which has been a subject of research in several fields[20]. While a fundamental understanding of this system is lacking, charge and polarity gradients are thought to play a role. Hence, it is prudent to perform control experiments to ensure that the chemical and structural integrity of the species under study are not compromised. In the Supplementary Information (Note 4), we report on measurements performed on dye molecules with different charge states, experiencing more than 100-fold longer diffusion times.

To understand the third effect, namely a higher number of bursts in an OFA, we performed Monte Carlo simulations to investigate the influence of the fluidic confinement on the diffusion behavior (see Methods). The outcome shows that the probability to return to the observation volume[21] is approximately 9 times larger in the quasi-2D geometry of an OFA with a channel height of 500 nm than for the case of an open solution. Indeed, we observe up to about 8 times more bursts in experimental measurements if we consider bursts with signal-to-noise ratio (SNR) of at least 2.5. More frequent passages allow for shorter measurement times at a given concentration although it could also lead to crowding if the concentration is not adjusted accordingly.

One might expect the stratified architecture of an OFA to also affect the distribution of the excitation laser intensity and thus the observation volume. To investigate this, we performed finite difference time-domain (FDTD) simulations. We find that the lateral extension of the observation volume is merely modified by a few percent in the OFA geometry (see Supplementary Note 5 for details). Furthermore, one might wonder if the presence of the water-glass interface changes the excited-state lifetime of the fluorophore. We, thus, measured the fluorescence lifetime with and without the antenna. The measured values of 4.05 ns and 3.98 ns in OFA and open solution, respectively, coincide within the measurement error of 0.1 ns, indicating that a geometry-induced lifetime change is absent or negligible.

We have shown that an OFA not only improves the collection efficiency of fluorescence emission but it also extends the diffusion time and thus the overall brightness. Both phenomena are highly desirable for a wide range of studies on (1) weakly emitting fluorophores, (2) fast events that would benefit from higher photon rates and thus shorter integration times, and (3) slow events that require longer observation times. We remark that while the full antenna effect relies on the realization of a thin liquid layer, it is possible to achieve some of its advantages close to the air interface in a thicker water reservoir. We elaborate on this arrangement in the Supplementary Information (Note 6). In the next section, we present a case study, where an OFA is used to investigate a range of dynamic processes in one and the same molecular system.

## Real-time dynamics of a Holliday junction monitored via smFRET

Holliday junctions (HJ) are four-way DNA junctions formed during DNA strand exchange of homologous duplexes and play a fundamental role in genetic recombination[22]. Nevertheless, important questions regarding their conformational and thermodynamics features remain unresolved. In aqueous solution, HJs can adopt two conformations with branches stacked in parallel or anti-parallel manners (see sketch in Fig. 3a). Fluctuations between these two conformations have been the subject of many studies[23–26]. In particular, it has been shown that an increase in $Mg^{2+}$ concentration slows down the transition between them. Moreover, it has been speculated that a square planar

configuration might exist as an intermediate state between stacked and unstacked conformational states at physiological conditions. However, this prediction has not been confirmed due to insufficient temporal resolution in FRET measurements. We, therefore, used this system for a case study in which both slow and fast dynamic processes could be dialed in one system.

We labeled HJs at strands $a$ and $b$ with Alexa488 as donor (green channel) and Atto647N as acceptor (red channel), respectively. The details of the sample preparation can be found in Methods and Supplementary Information. We carefully characterized the internal HJ dynamics during diffusion inside the OFA to ensure that the antenna does not perturb them (see Supplementary Note 7). Here, we used single-molecule multi-parameter fluorescence detection (sm-MFD) in a pulsed interleaved excitation (PIE) scheme[27] to achieve polarization-dependent time-correlated single-photon counting with high accuracy and precision. In this fashion, we extracted multiple parameters for each single-molecule event such as fluorescence lifetime $\tau_{D(A)}$ of the donor in the presence of an acceptor molecule, intensity-based FRET efficiency $E$ defined as the ratio between the fluorescence signals of the acceptor and donor molecules (see Methods), stoichiometry, correlation amplitudes, and anisotropy values simultaneously[27]. Moreover, calibrated polarization measurements let us conclude that the OFA arrangement yields $g = 1.2$, where $g$ denotes the ratio of perpendicular to parallel polarization for the detected light[27]. Single-molecule measurements yielded a steady-state anisotropy of $r_{ss} = 0.06$, indicating free rotation and therefore unrestricted movement of the HJ and of the dye (see Supplementary Note 8).

In Fig. 3b, we present the recorded count rates for the donor and acceptor channels in a two-dimensional MFD histogram as well as their corresponding 1D projections for experiments performed in open solution (blue) and with an OFA (red). We identify three fluorescent populations: acceptor only (AO), donor only (DO), and the doubly-labeled case of donor and acceptor (DA). We verified that the antenna enhances the average photon detection rate by a factor of 4 for both donor and acceptor dyes (see Fig. 3b) while the FRET efficiency remains unchanged (Fig. 3c, d). Such an achromatic enhancement in the collection efficiency over more than 150 nm is an important feature of the antenna design, which is not easy to achieve in microcavities or plasmonic nano-antennas.

To ensure that the OFA does not introduce systematic effects on the internal dynamics of the HJ, we examined the cross correlation of photons in the green and red channels (see Supplementary Information). The quantity $\tau_{relx}$ defined at half the amplitude of the bunching term can be used as a measure for the characteristic relaxation time between the FRET states (see Supplementary Note 7). In the absence of $Mg^{2+}$, HJ displays high relaxation rates from one stacked conformation to the other with $\tau_{relx} \approx 130\,\mu s$. Considering that we obtained the same $\tau_{relx}$ values with and without OFA, we conclude that the intrinsic dynamics of HJs are not affected in the OFA. For comparison, in Fig. 3c, d we plot the projections of the distributions for $E$ and $\tau_{D(A)}$ (see violet histogram curves) obtained for 4WJs[28] in the absence of $Mg^{2+}$ ions (see also Supplementary Fig. 8). In this case, the transitions between the two conformational states of HJ occur faster than the burst duration such that the different FRET states culminate in average values. We also display two kinds of FRET lines as graphical guidance[29,30]. Molecules with no conformational exchange during the burst duration lie on the static FRET line (black) while molecules that exhibit conformational exchange (dynamic molecules) are shifted to the dynamic FRET line (cyan).

## Slow HJ dynamics

To examine a system with slow dynamics, we added $Mg^{2+}$ ions, which are known to slow down the HJ transition to the millisecond regime[24]. The black FRET line in Fig. 3c depicts the behavior expected from a static HJ, and deviations of the FRET sm-populations from this curve

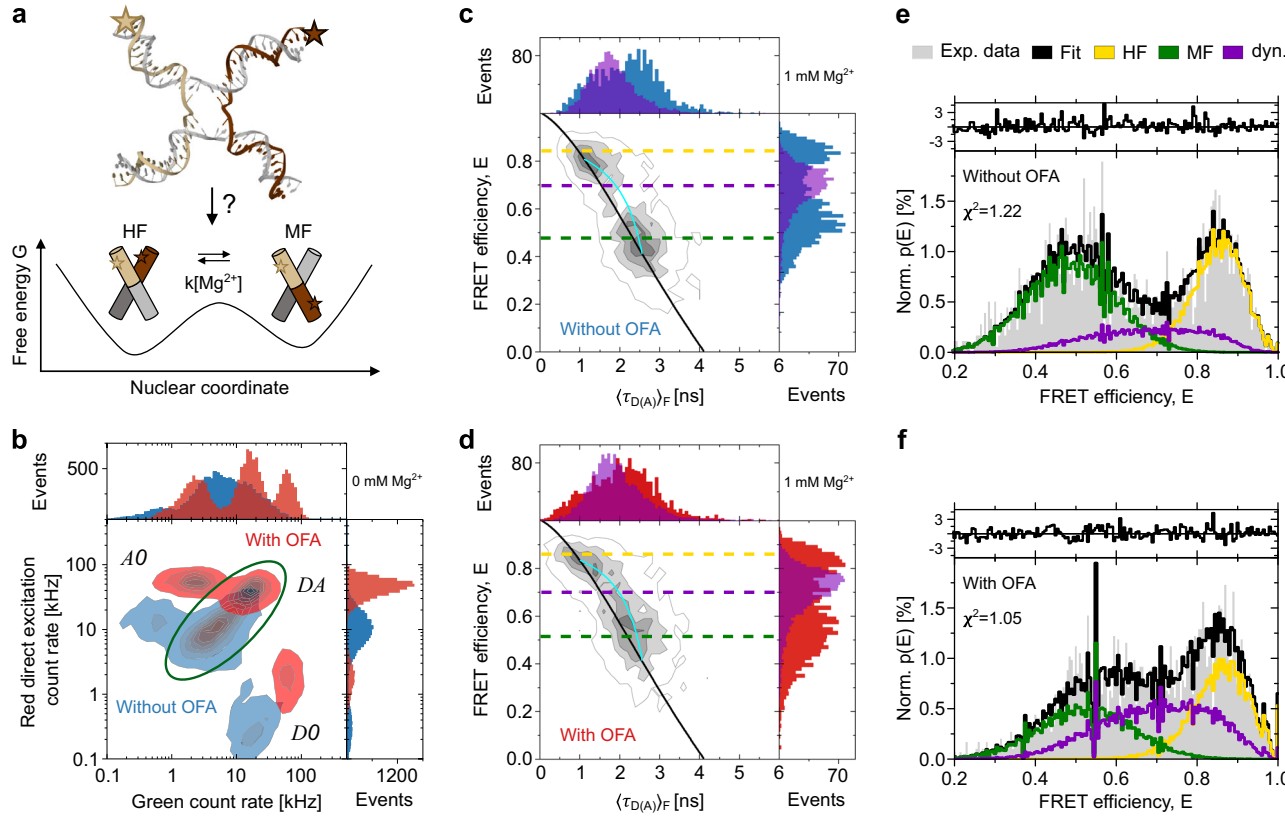

**Fig. 3 | Single-molecule multiparameter fluorescence detection (sm-MFD) on Holliday junctions. a** Sketch of the transition of a HJ between two stacked conformations assigned by high FRET (HF) and middle FRET (MF) states. Labeled arms of the junction are shown in cream (donor) and brown (acceptor). The rate $k[Mg^{2+}]$ between these two stable states is high without $Mg^{2+}$ and becomes lower as $Mg^{2+}$ is added. **b** Donor and acceptor count rates in direct excitation obtained from single-molecule bursts with two 1D projections. Only doubly-labeled molecules were selected using a stoichiometry and an ALEX-2CDE filter (see Supplementary Information). *DO*: donor only, *AO*: acceptor only, *DA* double-labeled. **c**, **d** sm-MFD measurements in a 1 mM $Mg^{2+}$ buffer without (**c**) and with (**d**) OFA. Only doubly-labeled molecules were selected for this analysis. Black lines represent static FRET. Dark yellow, violet, and dark green dashed lines indicate the center of the populations fitted in a PDA analysis (**e**, **f**). The OFA measurement shows more dynamical averaging in the region between HF and MF populations since the diffusion time is longer. For comparison, we display the dynamic 4WJ junctions (violet population in the 1D histograms in (**c**, **d**)) that result from a fast exchange in the absence of $Mg^{2+}$ (see Supplementary Figs. 8, 17). The dynamic FRET line between the two FRET species is shown in cyan. **e**, **f** Corresponding PDA with/without the OFA from (c,d). Histograms are based on a 2 ms time window, and the analysis was done globally for time windows of 1, 2 and 3 ms. A two-state model of HJ was applied where the dynamic fraction (violet line) between the HF (dark yellow line) and MF (dark green line) population was observed and extracted. Top panels in (**e**, **f**) display the residuals of the fit process to the experimental data. Resulting superposition of these 3 species is shown in black and are compared to experimental data in gray. The legend for figures (**e**, **f**) is shown in the upper right corner. The excitation power for these measurements was lowered to 60 μW to avoid photobleaching in the observation volume.

indicate a hallmark of conformational exchange dynamics[29]. Indeed, Fig. 3c shows that in a conventional measurement, one obtains two well-resolved populations at a $Mg^{2+}$ concentration of 1 mM. At this concentration, the conformation exchange is on the millisecond time scale. The data in Fig. 3d, however, show a considerably larger population between the two stable FRET states in an OFA (dashed violet line) that exhibit a shift to the dynamic FRET line. It is expected that molecules exhibit conformational exchange during their dwell time because the burst durations with OFA are longer than the average lifetime of one conformer. These results verify that extended observation times (i.e. improved recurrence analysis of single particles (RASP)[21]) and larger brightness in the antenna provide insight into slower dynamical processes than those accessible in bulk studies (see Supplementary Fig. 15). At the same time, the fact that the populations in Fig. 3c, d coincide provides further evidence that the intrinsic HJ dynamics are not affected by the OFA.

To confirm these observations and to increase the temporal resolution, we also performed dynamic Photon Distribution Analysis (dPDA)[28], where each burst is divided into equally-sized time windows (TW) with typical lengths of 1, 2, and 3 ms. Then $E$ is extracted from each TW and arranged in a histogram, whereby the leakage of the donor signal to the acceptor channel, dyes' fluorescence quantum yields, and detection efficiencies ratio of the donor and acceptor channels have to be considered for an accurate description of the FRET states. Finally, a fit model including the number of the FRET states is applied to the experimental data. Figure 3e presents the outcome for measurements without OFA, identifying two stable states of medium FRET (MF) and high FRET (HF) efficiency as well as a small population (violet curve) between these two, in agreement with the populations in Fig. 3c. Figure 3f shows the result of the same analysis with an OFA. In this case, it is visible that the dynamic population (violet curve) is considerably larger than before, while the stable FRET states remain unperturbed (green and yellow curves). We verified that this experimental observation can be reproduced with the same kinetic parameters in the simulated data if the diffusion time is increased from $\tau_{diff} = 0.7$ ms (no OFA), to $\tau_{diff} = 2$ ms (with OFA) to describe the longer dwell times in the OFA (see Supplementary Note 9).

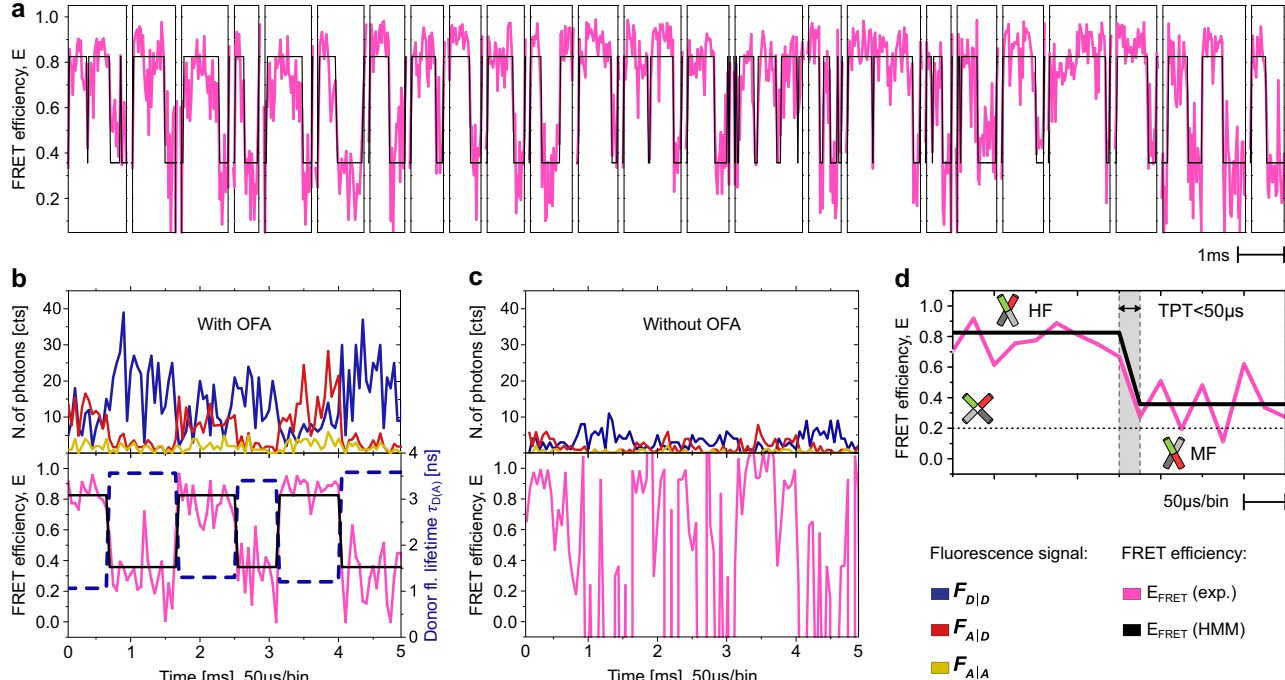

**Fig. 4 | HJ single-molecule burst trajectory. a** The pink curve plots FRET efficiency ($E$) traces obtained from individual bursts in an OFA with high excitation rate at $I_{exc(485)} = 800\,\mu W$ using a buffer containing 0.2 mM Mg$^{2+}$. Each burst is shown separately in the trace to emphasize that the trace is composed of bursts that are stitched together. Only bursts with a diffusion time higher than 0.8 ms that show a transition from HF to MF or vice versa were considered. Black lines present the outcome of a fit based on a hidden Markov modeling (HMM) algorithm. **b** Close-up of an $E$ trace in an OFA. Top panel displays the fluorescence intensities where $F_{D|D}$ corresponds to the fluorescence from the donor under donor excitation (dark blue trace); $F_{A|D}$ is the fluorescence of the acceptor under donor excitation (dark red trace), and $F_{A|A}$ is the fluorescence of the acceptor under acceptor excitation (dark yellow trace). The lower panel presents $E$, FRET levels obtained with the HMM analysis ($E_{FRET}$ HMM, solid black line, left axis) and the fluorescence lifetime of the donor molecule ($\tau_{D(A)}$, dashed dark blue trace, right axis). Lifetimes were fitted for the interval where a stable FRET level was identified. **c** Same trajectory as in (**b**) but with 75% fewer photons to mimic the absence of OFA. **d** Close-up of an $E$ trajectory using a 50 µs time binning. The transition always occurred within a single bin resulting in a transition path time (TPT) of less than 50 µs. The legend for figures (**b**, **c**, **d**) is presented in the lower right corner.

## Fast HJ dynamics

Previous measurements on immobilized biomolecules have raised the question of how fast the major conformational transitions can be and how rugged the corresponding transition paths are[31–34]. The high photon collection efficiency afforded by OFAs enables one to apply short binning times to single bursts in the experimental time trace, thus, visualizing the real-time dynamics in a contact-free environment. To suppress the inherent effects of photobleaching and blinking in fluorophores, we also added photo-stabilizers following the protocol of Ref. 35 (see "Experimental conditions" in Methods). To showcase the resulting improved temporal resolution, we performed measurements at lowered concentrations of Mg$^{2+}$ (0.2 and 0.5 mM). Figure 4a displays a FRET trajectory assembled from 50 individual bursts that have been selected from different positions in the original time trace. The resulting $E$ trajectory was analyzed with the software HaMMy[36] based on a hidden Markov model (HMM). Here, a static model was used to fit the data since FRET levels can still be mapped with several data points at binning times shorter than 100 µs. In fact, it follows that dynamical averaging can be avoided at an even shorter binning time of 50 µs (for more details, see Supplementary Note 10).

The results of the HMM analysis allow us to quantify the exponential decay of level durations for the HF and MF states directly from the trace, yielding relaxation times $\tau_{relx} = 250\,\mu s$ and $\tau_{relx} = 310\,\mu s$ for the 0.2 mM and 0.5 mM concentrations of Mg$^{2+}$ ions, respectively. These values agree with the general trend of HJ to have larger relaxation times for higher Mg$^{2+}$ concentration[37] and closely match the ones obtained with a filtered fluorescence correlation spectroscopy (fFCS) analysis (see Supplementary Note 7). We note that each burst in the composed $E$ trajectory might terminate due to translational diffusion out of the observation volume before the FRET level ends, shortening the relaxation times. In agreement with the results of Kalinin, et al., simulations indicate that $\tau_{relx}$ is shortened by roughly 30% when analyzing merged bursts of single-molecule events with HMM (see Supplementary Note 10)[28]. We complemented the $E$ trace with dPDA analysis for an accurate estimate of the relaxation rates.

Next, we studied trajectories directly at high temporal resolution in order to (1) estimate an upper bound for the transition path time of HJ to go from one conformer to the other, and (2) check whether an intermediate state is detectable. Figure 4b shows a close-up of trajectories of the two background-corrected fluorescence channels recorded at 1.3 MHz. A visual inspection of the raw data in the upper panel clearly reveals the alternate FRET efficiency level modulations. In the lower panel, we plot $E$ (magenta line, left vertical axis) and $\tau_{DA}$ (black line, right vertical axis) fitted for the interval of a stable FRET level, showing the expected anti-correlation. To demonstrate the advantage of the OFA, we mimicked bulk measurements with low collection efficiency by randomly deleting 75% of the detected photons from the experimentally recorded data. As seen in Fig. 4c, $E$ levels are no longer identifiable at a binning time of 50 µs. In the Supplementary Information (Note 10), we present a more complex analysis where simulated traces in an OFA and in a confocal single-molecule arrangement are compared for our experimental conditions.

To estimate the limit for detecting transitions in FRET efficiency traces as a function of the time window (tw) and the differences of FRET efficiency levels $\Delta E$, we compute the attainable SNR (see Supplementary Note 10 for more details). The simulations show that a broad distribution of $E$ ($\sigma_{eff} = 0.32$) without an OFA prevents the HMM algorithm from identifying reasonable FRET levels. In contrast, the

distribution is significantly narrower ($\sigma_{eff}$ = 0.14) with an OFA, enabling the HMM algorithm to find stable FRET levels efficiently. Moreover, we show in the Supplementary Information (Note 10) that under our experimental conditions, application of an OFA increases the experimental signal-to-noise ratio by a factor ~4.5. This corresponds to a temporal resolution of ~20 μs for SNR = 1. We note that high temporal resolutions have also been reported in conventional arrangements solely through the use of photo-stabilizers and saturation of fluorophores at high excitation powers[35,38,39], but an OFA offers a decisive advantage because it provides higher count rates at lower excitation powers and, thus, less photodamage.

In Fig. 4d, we plot a close-up of a FRET transition from HF to MF, whereby the horizontal dashed line signifies the expected $E$ value of the square planar intermediate calculated using the software FRET Positioning and Screening (FPS)[40]. Since we could not detect a stable third state on this time scale in the FRET trace, we conclude that the transition path time is shorter than 50 μs and that the lifetime of a potential intermediate state must be smaller than this value.

## Discussion

We have introduced and characterized an optofluidic antenna, which not only enhances the optical signal from emitters in a liquid, but it also provides longer diffusion and observation times, thus allowing more sensitive measurements. The fabrication of the device is inexpensive and straightforward (see Supplementary Note 1). Moreover, it operates in a broad spectral domain and is highly fault tolerant with regard to the antenna dimensions. A decisive advantage of our OFA design is that it can be readily implemented in existing inverted microscopes and is compatible with other microscopy methods such as dark-field and iSCAT microscopies beyond fluorescence, e.g., for nanoparticle analysis[41]. Moreover, it can also be combined with platforms such as plasmonic systems[10,42] or methods that slow down the translational diffusion of analytes such as trapping, immobilization, or tethering mechanisms[6,7,9]. These features usher in sensitive contact-free optical measurements that give access to both faster and slower dynamics of biological entities than are available in a regular bulk fluidic environment.

Measurements with an OFA allow one to expand the time range for studying biomolecular dynamics beyond the limit imposed by the translational diffusion time in a laser focus. Figure 4d demonstrates this for a transition within 50 μs with SNR = 1.5 while our analysis predicts SNR = 1 for measurements with a temporal resolution of 20 μs (see Supplementary Fig. 13d). On the slow side, we have measured up to about 1 ms, but application of the recurrence analysis of single particles (RASP) harbors the potential to reach tens of milliseconds[21]. In Supplementary Fig. 15, we show recurrence FRET efficiency plots without and with OFA, respectively, which demonstrate that measurements with OFA have more recurrence events and lower shot noise so that the time window for the analysis of very slow exchange kinetics is significantly expanded. In addition, more efficient collection of photons by about 2.2 folds in an OFA gives access to the optimal photon budget, which is fundamentally limited by photobleaching.

To demonstrate the OFA virtues on one single molecular system, we considered DNA four-way junctions with a molecular mass of about 100 kDa, comparable to the size of many proteins and biomolecular machinery used for single-molecule studies of protein folding and catalysis[43,44]. While we have focused on the physical performance of OFAs, it would be exciting to augment OFA measurements with more sophisticated analysis[33,45,46] to achieve even higher temporal resolution and sensitivity. The broad spectral bandwidth of the OFA and enhancement of the photon counts also advocate its use for accurate FRET-derived distance measurements at Ångstrom precision[3]. Moreover, the architecture of the OFA offers an ideal platform for future research on the effect of the water-air interface on molecular diffusion.

## Methods

### Antenna preparation

Cover glass substrates were purchased from Marienfeld-Superior (model No.1.5H-170 μm, tolerance 5 μm). Micropipettes were either prepared by heat-pulling glass capillaries (World Precision Instruments (model TW-100-3)) using a Sutter Instruments 388 Co (model P-2000) or ordered directly from Hilgenberg GmbH, Germany. The pipette can be specified differently. A possible choice is: micropipettes made of borosilicate glass, with a total length of (55 ± 5) mm, an outer diameter of (1.0 ± 0.05) mm, an inner diameter of (0.75 ± 0.05) mm, a taper length of about 10 mm, and a tip opening inner diameter of (0.02 ± 0.005) mm. In either case, the micropipettes were dipped in dichlorodimethylsilane to make their inner wall hydrophobic. The remaining dichlorodimethylsilane inside the micropipette was removed afterwards by dipping the micropipettes in acetone in an ultrasonic bath for several minutes. A further step of cleaning with isopropanol (IPA) for two minutes in ultrasonic bath removed residual acetone. Micropipettes were heated for several minutes at 250 °C to remove the remaining IPA.

### OFA assembly and multiparameter fluorescence detection (MFD) setup

The experimental assembly of the OFA is described in detail in the Supplementary Information (Note 1). Experiments were performed on a home-built sm-MFD setup using an inverted microscope with a pulsed interleaved excitation (PIE) scheme. Position of the antenna was calibrated using a CCD camera. Solutions were diluted to pM concentrations to select single molecule events only. Further details can be found in the Supplementary Information (Note 11).

### Fluorescence correlation spectroscopy (FCS)

Fluorescence correlation analysis was done using unfiltered signal for free dye studies and a fluorescence lifetime filtered signal for HJ measurements. We applied a model including diffusion in a 3D Gaussian shaped volume with a triplet state. Brightness was calculated using the number of molecules in a bright state $N_{bright}$ and fitted using an exponential saturation curve. More details can be found in the Supplementary Information (Notes 2, 7).

### Experimental conditions

The measurement buffer was ultrapure water for Rhodamin110. For HJ, the buffer contained 10 mM Tris, 50 mM NaCl at pH = 7.5 with a varying amount of Mg$^{2+}$ in the case of normal excitation ($I_{exc(485)}$ = 60 μW). At high excitation intensity ($I_{exc(485)}$ = 800 μW), additional photoprotection was used by adding 1 mM Trolox and 10 mM cysteamine as suggested in Ref. 35 to suppress the effects of photobleaching and photoblinking. We passivated the cover glass surface by applying a 10 μM concentrated BSA solution, following published protocols[47,48].

The lateral extension ($w_0$ = 1 μm) of the excitation/observation region is extracted from the calibration of the experimental setup, whereby $w_0$ is defined as the width, where the intensity decays to $1/e^2$ of its maximum. The pinhole size was set to coincide with the lateral extension of the beam width in the focal plane.

### Sample preparation

Labeled and unlabeled single strands were ordered from IBA-Lifesciences GmbH, Germany. Single strands had the following sequence: α (5'-CCT AAT TAC CAG TCC AGA TTA ATC AGT ACG), β (5'-CGT ACT GAT TAA TCT CCG CAA ATG TGA ACG), γ (5'- CGT TCA CAT TTG CGG TCT TCT ATC TCC ACG), δ (5'-CGT GGA GAT AGA AGA GGA CTG GTA ATT AGG). Strand α was labeled at nucleotide 7 (thymine) using Alexa488 and strand β at 10 (thymine) using Atto-647N, hence the hybridized sample is called D(a)A(b). Hybridization was performed using a PCR machine (Pico17, Thermo Fisher Scientific) at a concentration ratio of 1:3 (labeled to unlabeled) with 5 nM and 15 nM, respectively. The sample was heated up to 85 °C, quickly cooled down

to 52 °C, where the temperature was held for 2 h, and then cooled down to 4 °C. The protocol leads to a typical ratio >80% of doubly labeled molecules. The ratio was monitored using sm-MFD via stoichiometry value (see example in Supplementary Fig. 9). Anisotropy values were monitored to exclude the existence of free dye in the sample.

## Monte Carlo simulations of diffusion

We assume that the excitation power is sufficiently low that photo-bleaching can be neglected. The simulation starts with the analyte placed at the center of the observation volume, and $N = 10^7$ steps of diffusion are considered in a total time of 800 ms. The process was repeated $10^4$ times at steps of 80 ns sufficient to map the translational diffusion of the molecules. A burst is created for every entry-exit event through the observation volume (see Supplementary Note 12 for more details). The boundary surfaces of the OFA were set to steer the molecule in a random direction after each contact. The shape of the water meniscus at the air interface was approximated by an ellipse with a semi-minor axis length of 500 nm in the axial direction and a semi-major axis of 7 μm, corresponding to the dimensions of the micro-pipette. We verified that the simulation results are not sensitive to the exact shape of the interface but rather to the surface concavity[49].

## Molecular detection efficiency (MDE)

FCS and FRET measurements were performed on an inverted confocal microscope. Fluorescence excitation is confined to a small region defined by the focused laser beam. Transmission of the emitted photons to the detector depends on the numerical aperture of the objective lens and the size of the pinhole in the image plane. The combined effect of the optical system on the signal intensity is characterized by the molecular detection efficiency (MDE) calculated by modelling the excitation intensity $I(r,z)$ as a Gaussian beam and accounting for the image of a molecule located at $r$, i.e. the point-spread function $\mathrm{PSF}(r,r',z)$, using vectorial diffraction. MDE is then given by

$$\mathrm{MDE}(r,z) = I(r,z) \cdot \int \mathrm{PSF}(r,r',z) \cdot T(r')dr,$$

where $T(r')$ represents transmission through the pinhole. In order to assess the relative enhancement of photon detection inside an OFA, we compared the spatial averages of the MDE for varying water layer thickness.

## Single-molecule Multiparameter Fluorescence Detection (sm-MFD)

Single-molecule bursts were identified using a burst search algorithm according to Ref. 18 using a Lee filter, an inter-photon time threshold of typically 0.1 ms and a minimum of 60 photons per burst. Each burst was divided into equally sized TWs of a length of 200 μs for TW-based analysis and 50 μs for intensity-based trace analysis, respectively. Correction factors for intensity based confocal MFD were estimated using spectra of the used dyes and measured spectra of the optical components based on Ref. 3. We obtained the fluorescence signal by subtracting a background from the registered signal. Next, we determined several correction factors for crosstalk/donor leakage ($\alpha$), different excitation flux ($\beta$), and the ratio of detection efficiency and quantum yields ($\gamma$) to compute the fully corrected fluorescence signal as described in Eqs. (1–3). Fluorescence quantum yields were estimated using the lifetime of the donor-only population for the donor and the lifetime of the direct acceptor excitation for the acceptor. It follows for the fluorescence of the donor $F_D$ after donor excitation $F_{D|D}$ that

$$F_{D|D} = \gamma \cdot I_{D|D}^{ii}, \tag{1}$$

where $I_{D|D}^{ii}$ denotes the background corrected raw intensity of the donor after donor excitation. In the same way it follows for the

acceptor fluorescence after acceptor excitation ($F_{A|A}$) and after donor excitation ($F_{A|D}$):

$$F_{A|A} = \frac{1}{\beta} \cdot I_{A|A}^{ii}, \tag{2}$$

$$F_{A|D} = I_{A|D}^{ii} - \alpha I_{D|D}^{ii} - \delta I_{A|A}^{ii}. \tag{3}$$

From these, one can estimate the FRET efficiency $E$ and the stoichiometry $S$ as follows:

$$E = \frac{F_{A|D}}{F_{A|D} + F_{D|D}}, \tag{4}$$

$$S = \frac{F_{D|D} + F_{A|D}}{F_{D|D} + F_{A|D}F_{A|D}}, \tag{5}$$

In this work $S$ was used to monitor the labeling quality and hybridization efficiency of HJs (see Supplementary Note 8).

## FRET efficiency trajectory

FRET efficiency trajectories were assembled from single bursts taken under high excitation rate. Bursts were filtered using the sm-MFD approach based on their stoichiometry, ALEX-2CDE value[50] and differences in macrotimes to filter out bleached molecules. Additionally, a minimum diffusion time of $t_d > 0.8$ ms was applied as threshold since the majority of bursts were bleached after 0.2 ms due to the high excitation rate.

## Hidden Markov model

Fitting of the FRET efficiency trajectories was done using an HMM with theory and software from Ref. 36. HMM is suitable to find FRET efficiency levels hidden in noise in a trajectory based on time-binned data. The model is applicable for a multi-state system where each single state decays exponentially. It describes these systems with transition probabilities for the sample between one state and another and emission probabilities that model the distribution of $E$ values with Gaussian distributions.

## Data simulation

Simulated data were generated using a Brownian dynamics approach with in-house software available upon request[51]. Molecules were simulated according to the experimental $E$ levels, brightness, dynamic rate and diffusion time. For the diffusion of the molecule we considered free diffusion in a 3D Gaussian volume.

## FRET lines

Static FRET lines were computed following the methodology developed by Kalinin and Opanasyuk[28,29]. In short, the static FRET efficiency line was calculated as the ratio of the donor lifetime in presence of an acceptor $\tau_{D(A)}$ and in absence of an acceptor $\tau_{D(0)}$:

$$E_{\mathrm{static}} = 1 - \frac{\langle \tau_{D(A)} \rangle_f}{\tau_{D(0)}}. \tag{6}$$

Molecules show a distribution in the static FRET line corrected for linker dynamics if no dynamic averaging takes place in the signal. In contrast, the measured photon weighted average fluorescence lifetime $\langle \tau_{D(A)} \rangle_F$ is biased towards longer lifetimes if dynamic averaging occurs. This shifts the population off the static line.

## Photon distribution analysis (PDA)

In addition to the burst-wise approach, we also implemented a TW approach. Here, we used procedures from Refs. 28,52. The size of the

chosen TWs was 1, 2 and 3 ms for normal excitation rate experiments and 50 μs for high excitation rates. Histograms for FRET related parameters calculated for each TW ($E$ values or ratios of the green and red signals $S_{green}/S_{red}$) were binned, typically with $N_{bin} = 81$. Histograms were analyzed using a two-state model in the case of HJs (see Fig. 3a) with dynamic rates $k_{12}$ for HF to MF and $k_{21}$ for MF to HF. A standard Levenberg-Marquardt algorithm was used for the fitting procdure.

## Data availability

All relevant data supporting the key findings of this study, this is, raw (Picoquant TTTR format) and processed data as well as the analysis tools are available in the Zenodo database under accession code https://doi.org/10.5281/zenodo.10575477. Source data are provided with this paper.

## Code availability

The custom software and programs generated for the calculations in this study are available upon request. In-house programs are used (1) in the confocal multiparameter fluorescence detection experiments, (2) to elucidate the filtered fluorescence correlation spectroscopy curves, and (3) to analyze the fluorescence lifetime measurements. Software for analysis of single-molecule measurements and fluorescence correlation analysis and their simulation is available at http://www.mpc.uniduesseldorf.de and software for analysis of fluorescence decays can be downloaded from http://www.fret.at/tutorial/chisurf/.

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

## Acknowledgements
V.S. acknowledges fruitful discussions with Gilad Haran, Mischa Bonn and Hisham Mazal. This research in Erlangen was funded by the Max Planck Society (V.S.). The Düsseldorf group was funded by the Deutsche Forschungsgemeinschaft (DFG, German Research Foundation) grant SE 1195/17-1 and CRC 1208 (project A08) as well as the European Research Council through the Advanced Grant 2014 hybridFRET (671208) to C.A.M.S. J.F. acknowledges the support of the International Helmholtz Research School of Biophysics and Soft Matter (BioSoft).

## Author contributions
L.M-I. built the OFA setup and performed its basic characterizations. F-F.W. and L.M-I. performed analytical calculations of the intensity distributions and Monte Carlo simulations of the molecular diffusion. L.M-I., J.F., S.F. and R.K. performed and analyzed the FCS and smFRET measurements. V.S. conceived the experiment. S.G., C.A.M.S. and V.S. supervised the project. L.M-I., J.F., C.A.M.S., S.G. and V.S. wrote the manuscript. All authors discussed the results and revised the manuscript.

## Funding

## Competing interests
L. M., S.G., and V.S. have applied for a PCT application, published as WO2022/207073. The remaining authors declare no competing interests.

## Additional information
**Supplementary information** The online version contains Supplementary Material available at https://doi.org/10.1038/s41467-024-46730-w.

