## [Peer Review File · Nature Communications]

An optofluidic antenna for enhancing the sensitivity of single-emitter measurementsREVIEWER COMMENTS

Reviewer #1 (Remarks to the Author):

Morales-Inostroza and coworkers propose an optofluidic antenna comprised of a micro pipette immersed into the imaging solution to form a micro cavity that enhances the emission from individual emitters, thereby bridging the gap between surface based smFRET measurements from which long, dynamics' trajectories can be derived (but are hard to implement) and confocal, or solution based smFRET measurements that are far much easier to implement, but provide static smFRET measurements. The solution is clever, and from my perspective is easier to implement other existing alternatives (e.g., nano cavities - but not necessarily surface-based assays). The manuscript is very well written and the experiments are robust. However, before recommending publication, I would like to following concerns to be addressed:

1, authors mention that the proposed system bridges the gap between fast and slow dynamics which is hard to justify using the presented data. Although fast (micro to nano second dynamics) are presented in the manuscript, slow (millisecond to second dynamics) are not. Indeed, the longest trajectory presented is about 5 ms in length (or at least this is how it appears to be). This temporal span is significantly shorter than that offered by surface-based smFRET which ranges between a few seconds to a few minutes (and whose intensity is also enhanced by the TIR evanescent field). In light of this, the authors may want to revisit their claim about bridging fast and slow dynamics, or to make it quantify early on how much slow and fast their method can resolve. Could the authors also comment on the error they obtain in their FRET measurements in 4b and benchmark this against the error obtained in other studies (e.g., from the lab of Scott Blanchard at Cornell) using surface-based smFRET.

2, the advantage of performing confocal based smFRET measurements is their ease - users literally pipette their solution containing the labelled molecules, switch on the microscope and collect data. With existing commercial as well as open-source solution [see smfbox; Ambrose et al. Nature Comms, 2020] such experiments are easy to implement compared to surface-based smFRET. Now, in the presence of optofluidic antenna, these confocal based

experiments will be much harder to implement. The antenna will require a glass pipette to be tapered, and to be installed on a piezo actuator (which would be only available to groups actively building optical microscopes) and for this piezo to be approached to the sample and for the gap length to be optimised through an automated calibration routine. Although an assembly is described in the SI, it is not clear how can a user who wants to implement this solution do so reproducibly. In addition, figure 1 in the SI needs to be clarified (or better expanded) to serve the above purpose.

Reviewer #2 (Remarks to the Author):

In this manuscript, the authors implemented a planar optofluidic antenna (OFA) with a stratified structure in fluidic arrangements. They used modelling and FCS experiments to carefully characterize the benefits of utilizing this OFA over the conventional open solution case in in-solution single-emitter measurements: enhanced detected signal, longer bursts for a series of dye molecules, and increased number of recorded bursts per unit time at a given concentration. They also did a case study on the FRET of Holliday junction using OFA to reveal the real-time conformational dynamics that is otherwise not well resolved by conventional measurements. The novel design and engineering of the OFA is the highlight of this work, which is described and illustrated in detail. The analyses of measurements are solid and logic. This work is not only inspiring for method developments, but also contributes new physical insights to single-molecule studies. I believe it could potentially raise interests to a broad range of the readers of the journal, especially those working in the biophysical field. I did not find major flaws in the manuscript. However, listed below are several minor comments that I suggest the authors to address before publication:

1. Figure 1c. I am confused why the blue (open solution) curve looks linear. According to Figure 1b the photon emission distribution as a function of the collection angle for the open solution case is constant, then the fraction of total photon collection calculated by integrating the solid angle should be $1/2(1-\cos \theta)$, where θ is the collection angle. Accordingly, the blue curve should be non-linear, and the photon collection efficiency enhancement of OFA is slightly larger than 2.2-fold.

2. In Figure 1a the water layer thickness in the middle is ~ 500 nm while the distance between the micropipette wall and the cover glass is only 100 nm. However, when modelling the distribution of the excitation light in the antenna in SI section B, the water layer thickness is a constant 500 nm. There is a substantial difference in the stratified geometry, although I expect the difference will not affect the results significantly. Nevertheless, the authors should mention and briefly discuss the difference about the shape of water layer in SI section B.

3. Figure 2g. The authors may need to do some normalization to the distributions to unambiguously show N_{tot} per burst is significantly larger in an OFA.

4. Page 4, line 116. When authors compare the FCS results, they list four effects: “1) the number of photons per unit of time is higher, 2) the average brightness is larger ...” What is the explicit difference of 2) from 1)?

5. Figure 3a. The horizontal and vertical axes are not marked.

6. Figure 3. What is exactly the physical nature of the “dynamic fraction” depicted by violet? Resolving its population is the highlight of the sm-MFD study on Holliday junctions, yet the discussion of the “dynamic fraction” is a little vague. Suppose within this state the switch happens frequently as the name suggests, why is the corresponding cluster not further deviating from the static FRET line in Figure 3d compared to those of the HF and MF states? And why is the corresponding FRET efficiency spans much larger than those of the HF and MF states in Figure 3e,f?

7. Page 5, line 133. The authors performed Monte Carlo simulations and found “the probability to return to the observation volume is approximately 9 times larger in the quasi-2D geometry of an OFA with a channel height of 500 nm than for the case of an open solution”. The illustration from SI section C seems to imply that the recurrence is largely due to the spatial confinement in the z direction since the distance between the micropipette wall and the cover glass is only 100 nm. It seems that, for relatively large biomolecules, the

signals in FCS experiments could mostly come from the same molecules within a very small OFA solution volume. Therefore, the emitters in OFA may not be in free diffusion like those in the open solution case. It would be good if the authors can comment on this and the effect of the spatial confinement of OFA on sampling of single emitters in solution.

A follow-up on the free diffusion problem. It seems the water meniscus has a large contact area with air, so evaporation is probably unavoidable. Then it is likely that the solution continuously flows into the OFA to maintain the shape of the water meniscus, which also suggests that the molecules measured may not be in free diffusion. In SI section A the authors briefly mentioned controlling the pressure inside the micropipette by tuning the syringe plunger. It would be better if the authors can give more details, e.g., what pressure is desired to maintain the current shape of the water meniscus? Also, how will the pressure applied affect water evaporation within the OFA?

8. Page 6, line 179. The authors claimed, “the antenna enhances the average photon detection rate by a factor of 4 for both donor and acceptor dyes while their ratio remains unchanged”. How was the quantitative conclusion that the ratio remains unchanged from Figure 3b made? By comparing the linear regression fittings of the two DA clusters?

9. SI section M. Comparing Figure 13a,e it seems that when decreasing the Mg^{2+} concentration from 0.5 mM to 0.2 mM, the HF state population increases substantially. It is well-known that Mg^{2+} ions slow down the Holliday junction transitions, but why do they also alter the FRET state populations here? Do authors have an explanation on the data shown in Figure 13?

Reviewer #3 (Remarks to the Author):

This manuscript describes an interesting technical advance in single emitter/molecule measurements using an optofluidic antenna. Single molecule fluorescence measurements are powerful tools to extract distributions and dynamics in complex systems in biology and other scientific areas. While well established in many forms, the methods still have limitations. Along these lines, the authors devise an optofluidic antenna (combined with smMFD/PIE) that can result in multiple improvements to such measurements, including

enhancements per passage in fluorescence signal (by improvements in collection efficiency and trajectory positioning in detection region) and diffusion time, as well as number of passages per molecule. The improvements are nicely demonstrated using the dynamics of a well-studied model system, the DNA 4-way junction. The work is carefully performed and described. The new method and capabilities will be interesting and useful for the scientific community. However, I have a few issues that should be addressed.

From the point of view of application of the method, one potential issue is the presence of large interfacial areas in the detection region. As is well established in the field (and noted in the manuscript), proteins and other biopolymers can have interactions with such interfaces, which can result in substantial perturbations to the properties of the system. While addressed somewhat, I think it is important for the authors to address this issue in more detail, and possibly suggest directions that may be used to alleviate it in future studies of more surface-sensitive systems.

The interpretation of the purple populations in Figure 3 in terms of the structural model were unclear - authors please explain further. Also, why is this population higher for the OFA vs open geometries - is it intrinsic or related to surface effects?

It is possible that the 4wj accesses a distribution of states in transitions between the stable ones. How would this be handled in the analysis of higher time-resolution traces and can any insight be gained for such a situation?

Since this is a major consideration for implementation of the OFA, it would be useful to include a little more detail of how the micropipette tip is positioned/held at the sub-micron distances from the coverglass surface (specifically how the absolute distance is measured/maintained).

In the section on 4wj slow dynamics, the authors note "These results verify that extended observation times and larger brightness in the antenna provide insight into slower dynamical processes than those accessible in bulk studies". This statement is somewhat unclear - what additional information about dynamics was obtained here with the OFA?

Minor:

50 microsecond binning times have been achieved previously in smFRET measurements using other techniques, e.g., doi.org/10.1021/ja9027023 or doi.org/10.1038/nmeth.1553

The authors may choose to include a couple of additional references in this regard.

Is it feasible to perform RASP-type analysis for the slow 4wj experiments?

There appear to be missing error estimates for some of the values presented. Authors, please check and add as needed.

Response letter to the referees' comments

Nature Communications manuscript NCOMMS-23-03690-T, An optofluidic antenna for enhancing the sensitivity of single-emitter measurements

We thank the reviewers for the time and effort for formulating their detailed assessments of our manuscript. The comments have provided us with thoughtful and constructive comments. We also appreciate their overall positive view of our work.

We have carefully considered and addressed the reviewers' feedback. The revised text passages are highlighted in the manuscript and reproduced in this letter for the reviewers' convenience.

In the response letter, we use the following color scheme:

Blue: revised passages in the manuscript and Supplementary Information

Reviewer #1 (Remarks to the Author):

Comment R1.1: Morales-Inostroza and coworkers propose an optofluidic antenna comprised of a micro pipette immersed into the imaging solution to form a micro cavity that enhances the emission from individual emitters, thereby bridging the gap between surface based smFRET measurements from which long, dynamics' trajectories can be derived (but are hard to implement) and confocal, or solution based smFRET measurements that are far much easier to implement, but provide static smFRET measurements. The solution is clever, and from my perspective is easier to implement other existing alternatives (e.g., nano cavities - but not necessarily surface-based assays). The manuscript is very well written and the experiments are robust. However, before recommending publication, I would like to following concerns to be addressed:

Authors mention that the proposed system bridges the gap between fast and slow dynamics which is hard to justify using the presented data. Although fast (micro to nano second dynamics) are presented in the manuscript, slow (millisecond to second dynamics) are not. Indeed, the longest trajectory presented is about 5 ms in length (or at least this is how it appears to be). This temporal span is significantly shorter than that offered by surface-based smFRET which ranges between a few seconds to a few minutes (and whose intensity is also enhanced by the TIR evanescent field). In light of this, the authors may want to revisit their claim about bridging fast and slow dynamics, or to make it quantify early on how much slow and fast their method can resolve. Could the authors also comment on the error they obtain in their FRET measurements in 4b and benchmark this against the error obtained in other studies (e.g., from the lab of Scott Blanchard at Cornell) using surface-based smFRET.

Answer R1.1: We thank the referee for their careful consideration of our work and their generally positive assessment of its novelty and importance. Below, we address the individual points.

By "slow dynamics" we meant to refer in relation to the translation diffusion time of the sample, which is for our type of experiments around a few milliseconds. We note that we specify this in our abstract by writing "... both the slow (ms) and fast (50 us)..." We have now modified our formulation to avoid any

confusion with respect to much larger timescales achieved in other methods, e.g., those using immobilized samples. We write in the discussion and outlook section:

Measurements with an OFA allow one to expand the time range for studying biomolecular dynamics beyond the limit imposed by the translational diffusion time in a laser focus. Figure 4d demonstrates this for a transition within 50 μs with SNR=1.5 while our analysis predicts SNR=1 for measurements with a temporal resolution of 20 μs (see Supplementary Fig. 14d). On the slow side, we have measured up to about 1 ms, but application of the recurrence analysis of single particles (RASP) harbors the potential to reach tens of milliseconds [21]. In addition, more efficient collection of photons by about 2.2 folds in an OFA gives access to the optimal photon budget, which is fundamentally limited by photobleaching.

Additionally, we clarified the advantages of the OFA over conventional evanescent-field detection schemes, by adding the following sentence to the manuscript:

However, those approaches can maximally collect up to 40% of the radiation from a randomly oriented dipole and require the emitter to be within $\lambda/2\pi$ of the interface [13].

Finally, in order to provide more information on the noise analysis, we added a new section in the Supplementary Information, including the new Figure 14 to Supplementary Note 13.

To analyze the noise using different time binning of the FRET efficiency trajectory, we followed two approaches. The first is based on the standard deviation of the FRET efficiency, $\sigma(E)$, fitted to the signal by the hidden Markov model (HMM), which was analyzed using the software Hammy [36] (see Supplementary Fig. 14 a). The second method for determining $\sigma(E)$ is based on a statistical noise analysis of the total mean number of photons that were obtained using different time bins. To do so, the following equation was applied

$$\sigma(E) = \sqrt{\frac{1}{N_F} E \left(E \left(1 - \frac{1}{\gamma} \right) + \frac{1}{\gamma} \right) \sqrt{\frac{1-E}{E}}},$$

where N_F is the total mean number of donor and acceptor photons for every time resolution, E is the FRET efficiency and $1/\gamma$ is the inverse normalized photon collection yield with $1/\gamma = (\Psi_D \cdot \Phi_{F,D(0)}) / (\Psi_A \cdot \Phi_{F,A})$, which is based on the experimental detection efficiencies of the donor and acceptor ($\Psi_D = 0.8$, $\Psi_A = 1$) and their effective fluorescence quantum yields ($\Phi_{F,D(0)} = 0.8$, $\Phi_{F,A} = 0.23$) (see Supplementary Fig. 14 b).

In eq. 13, we define a signal-to-noise ratio (SNR) of a FRET level as

$$SNR = \frac{\Delta E}{\sigma(E_1) + \sigma(E_2)} = \frac{\Delta E}{\sigma_{total}(E_1, E_2)}$$

where SNR = 1 corresponds to the threshold for resolving two distinct FRET efficiency levels. For SNR values > 1 , the experimental noise is small enough to resolve the two FRET efficiency levels for a given FRET contrast $\Delta E = E_2 - E_1$. Using our measured FRET efficiency levels ($E_2 = 0.81$ and $E_1 = 0.35$, i.e., $\Delta E = 0.46$), the resulting SNR computed by both methods are depicted in Supplementary Fig. 14 c. In Supplementary Fig. 14 d, the SNR with and without OFA are compared. It can be seen that with OFA we achieve a time resolution of $\sim 20 \mu\text{s}$, whereas the observation times without OFA are much longer ($\sim 90 \mu\text{s}$) even though

our FRET contrast of $\Delta E = 0.46$ is very high. In conclusion, under our experimental conditions, the temporal resolution is increased by a factor ~ 4.5 in an OFA.

Supplementary Fig. 14. **Noise analysis of a FRET efficiency trajectory of the HJ.** **a**, Standard deviation of the FRET efficiency for the two distinct FRET states of the HJ in dependence on the time resolution obtained by the Hidden-Markov Model. **b**, Standard deviation of the FRET efficiency for the two distinct FRET states of the HJ as a function of the mean number of photons N_F contained in individual bursts. **c**, Resulting SNR for the HMM (black line) and the mean number of photons N_F (magenta line) using eq. 12 and 13. **d**, Difference in the SNR using the OFA (red) and not using the OFA (blue). The black dotted horizontal line indicates the SNR threshold of 1.

We emphasize that we have benchmarked the performance of our approach against that of conventional confocal measurements in bulk liquid (see Fig. 2 of the main manuscript). We believe a comparison with methods that achieve long measurement times by immobilizing proteins is not very fruitful because these offer a different category of studies with different advantages and disadvantages as compared to freely diffusing proteins. Nevertheless, we have provided the interested reader with enough quantitative information to do this him/herself. We mention this result also in the main text by adding the following summary:

Moreover, we show in Supplementary Note 13 that under our experimental conditions, application of an OFA increases the experimental signal-to-noise ratio by a factor ~ 4.5 . This corresponds to a temporal resolution of $\sim 20 \mu\text{s}$ for $\text{SNR}=1$.

Comment R1.2: The advantage of performing confocal based smFRET measurements is their ease - users literally pipette their solution containing the labelled molecules, switch on the microscope and collect data. With existing commercial as well as open-source solution [see smfbox; Ambrose et al. Nature Comms, 2020] such experiments are easy to implement compared to surface-based smFRET. Now, in the presence of optofluidic antenna, these confocal based experiments will be much harder to implement. The antenna will require a glass pipette to be tapered, and to be installed on a piezo actuator (which would be only available to groups actively building optical microscopes) and for this piezo to be approached to the sample and for the gap length to be optimised through an automated calibration routine. Although an assembly is described in the SI, it is not clear how can a user who wants to implement this solution do so reproducibly. In addition, figure 1 in the SI needs to be clarified (or better expanded) to serve the above purpose.

Answer R1.2: We agree with the referee that the advantages of the OFA are accompanied by more sophisticated instrumentation as compared to the simplest implementation of confocal FCS. However, the assembly of an adjustable micropipette is straightforward and fully within reach of any scientific lab. Here are some of the main points to be kept in mind:

- 1- The experiments do not need custom-designed micropipettes. We now state on two occasions that we also use *commercially available* tapered micropipettes. For example, we have added the following text to the manuscript:
For the experiments reported in this work, we used both commercially available micropipettes and those heat pulled in our laboratory.
- 2- The components for the assembly, translation, and alignment of the micropipette are all available from several vendors such as ThorLabs or Newport. To make it easier for others to reproduce our OFA, we have added a more detailed description of the assembly and a new Figure 1 in Supplementary Note 1.
- 3- The antenna parameters are not very sensitive because its optical design principles are quite robust against variations. In other words, positioning of the micropipette tip does not need nanometer accuracy or precision although the mechanics offers that.
- 4- We tested the ease of operation of our manipulation stage by sending it from Erlangen to Düsseldorf, where it was mounted on a commercial microscope.
- 5- We note that the implementation of an OFA is by far not as complex as the procedure in other approaches such as an ABEL trap, which has nevertheless proven to become a valuable tool in the community. We believe the added value of measurements with OFA out-weigh the required technicality.
- 6- We anticipate that the pipette stage will become commercially available soon after the publication of our paper.

To facilitate the operation of an OFA setup, we have made some further changes in the manuscript and SI: In the Methods section, we have added the following text:

Micropipettes were either prepared by heat-pulling glass capillaries (World Precision Instruments (model TW-100-3)) using a Sutter Instruments 388 Co (model P-2000) or ordered directly from Hilgenberg GmbH, Germany. The pipette can be specified differently. A possible choice is: micropipettes made of borosilicate glass, with a total length of (55 ± 5) mm, an outer diameter of $(1,0 \pm 0,05)$ mm, an inner diameter of $(0,75 \pm 0,05)$ mm, a taper length of about 10 mm, and a tip opening inner diameter of $(0,02 \pm 0,005)$ mm.

We have also added the following text in Supplementary Note 1:

“After the coarse alignment described above is completed, the gap between the cover glass and the micropipette is filled by adding a blank buffer solution next to the micropipette (see sketch in Supplementary Fig. 1 e). The fine alignment of the angle between the micropipette end and the cover glass is adjusted by monitoring the interference fringes formed in the thin water layer. During this procedure, the OFA is illuminated with laser light (532 nm) in wide-field configuration.

Supplementary Fig. 2a-d shows the interference fringes formed as the angle between the micropipette end and the cover glass is corrected. When the micropipette end is positioned parallel to the cover glass, interference fringes have ideally vanished. Once the angle between the micropipette end and the cover glass is corrected, the thickness of the water layer can be adjusted via white light interferometry. The OFA is illuminated with a broad white light source. Supplementary Fig. 2e shows interference patterns for several thicknesses of the water layer. These interference patterns are recorded with a spectrometer implemented in the detection path. As the water layer thickness is reduced, the period of the white light interference is increased (see the figure caption for details). A similar procedure could also be implemented using a monochromatic light source such as an LED.

We note that we do not observe any change in the shape of the water meniscus within the duration of the measurements, which take typically several hours, i.e., as long as the macroscopic drop of buffer solution next to the micropipette is not evaporated. Therefore, we can exclude significant evaporation of the solution via the capillary. This process is also unlikely to influence the meniscus shape since the volume in the thin and long capillary is small and will quickly saturate above the solution.

We also want to stress that the OFA can be assembled rather fast due to the reproducibility of the positioning system depicted in Supplementary Fig. 1a. A simple flashlight from the top allows for the assembling of the OFA by observing the shadow of the micropipette. Supplementary Figs. 2f and 2g show two images of independent experiments where a micropipette enters the field of view. Next, the micropipette is positioned at the center of the field of view (see Supplementary Fig. 1d). Moreover, at this point, the micropipette end and the cover glass are brought in contact. This step is visually monitored by looking at the displacement in the z-direction of the focus spot formed in the center of the micropipette. Afterward, the distance between the micropipette and the cover glass is tuned using a piezo actuator to displace the micropipette in the z-direction.”

Reviewer #2 (Remarks to the Author):

In this manuscript, the authors implemented a planar optofluidic antenna (OFA) with a stratified structure in fluidic arrangements. They used modelling and FCS experiments to carefully characterize the benefits

of utilizing this OFA over the conventional open solution case in in-solution single-emitter measurements: enhanced detected signal, longer bursts for a series of dye molecules, and increased number of recorded bursts per unit time at a given concentration. They also did a case study on the FRET of Holliday junction using OFA to reveal the real-time conformational dynamics that is otherwise not well resolved by conventional measurements. The novel design and engineering of the OFA is the highlight of this work, which is described and illustrated in detail. The analyses of measurements are solid and logic. This work is not only inspiring for method developments, but also contributes new physical insights to single-molecule studies. I believe it could potentially raise interests to a broad range of the readers of the journal, especially those working in the biophysical field. I did not find major flaws in the manuscript. However, listed below are several minor comments that I suggest the authors to address before publication.

Answer R2: We thank the referee for appreciating the value of our work and for their constructive comments, which have helped us improve the manuscript. Below we address the individual points.

Comment R2.1: Figure 1c. I am confused why the blue (open solution) curve looks linear. According to Figure 1b the photon emission distribution as a function of the collection angle for the open solution case is constant, then the fraction of total photon collection calculated by integrating the solid angle should be $1/2(1-\cos \theta)$, where θ is the collection angle. Accordingly, the blue curve should be non-linear, and the photon collection efficiency enhancement of OFA is slightly larger than 2.2-fold.

Answer R2.1: We thank the reviewer for posing this question. The referee's expectation would indeed be correct if one were to use θ as a measure for the solid angle encompassed by it. In our representation, only a 2D cut along θ was shown, i.e., without integrating over ϕ . We have now decided to change the figure to show the more intuitive version that is alluded to by the referee. The figure below shows the outcome, which we include in the manuscript.

Figure R1. Calculated emitted power density (shaded curves) and collection efficiency (solid lines) averaged for a randomly oriented dipole as a function of the collection angle inside the OFA (red) and in an extended solution (blue). The vertical black dashed line indicates the maximum collection angle of the microscope objective.

The following lines have been added to the main text account for the changes in the figure:

The shaded curves in Fig. 1b display the normalized power density as a function of the opening angle of the collection optics, and Fig. 1c shows the fraction of the total number of emitted photons detected up to a certain collection angle. The insets in Fig 1c sketch the radiation pattern of a dipole averaged over all orientations (i.e., assuming fast molecular rotation) in a homogeneous open solution (blue) and in an OFA (red). We find that an average photon collection efficiency as high as 86% can be achieved for a fast rotating dipole in an OFA corresponding to nearly 2.2 fold more photons than in open solution.

Additionally, the following text was added to the caption of Fig1.

The insets show the respective radiation patterns of a dipole averaged over all orientations in a polar representation

Moreover, Supplementary Note 7 has been changed for a better description of our 2D representation (cut along θ). The following passages have been added:

... Supplementary Fig. 8 a shows a cut through the radiation pattern...

... Supplementary Fig. 8 e shows the radiated power in the y-z plane as a function of the collection angle
...

... Supplementary Fig. 8 f displays the corresponding photon collection efficiencies in the y-z plane again as a function of the collection angle (labeled with the same color code as in (e)).

Moreover, the following sentences were added to the figure caption of Supplementary Fig. 8:

... Cut along the y-z plane to display the radiation pattern in an open solution (a,) ...

... e, Radiated power in the y-z plane for the arrangements shown in a to d, and for the case of an emitter at 1000 nm from the air-water interface as a function of the collection angle...

... f) Calculated photon collection efficiency in the y-z plane. ...

Regarding the exact enhancement factor, we have changed the statement of 2.2 to “slightly larger than 2.2”.

Comment R2.2: In Figure 1a, the water layer thickness in the middle is ~500 nm while the distance between the micropipette wall and the cover glass is only 100 nm. However, when modelling the distribution of the excitation light in the antenna in SI section B, the water layer thickness is a constant 500 nm. There is a substantial difference in the stratified geometry, although I expect the difference will not affect the results significantly. Nevertheless, the authors should mention and briefly discuss the difference about the shape of water layer in SI section B.

Answer R2.2: The radius curvature of the water meniscus is much larger than the diameter of the laser focus so that it may be considered to be constant over the field of view. In a very conservative estimation, one can assume that the meniscus is approximated by an ellipsoid with a 500 nm short axis (the thickness of the water layer) and two 10 μm long axes. In such an arrangement, the thickness of the water layer decreases by less than 1 nm over the diameter of the laser spot. We included the following paragraph in Supplementary Note 4:

We note that in all numerical simulations, the water-air interface is assumed to be flat in the region of the observation volume. This is justified because the height of the water layer remains constant over the extension of the entire laser spot to within a few nanometers.

Comment R2.3: Figure 2g. The authors may need to do some normalization to the distributions to unambiguously show N_{tot} per burst is significantly larger in an OFA.

Answer R2.3: We thank the reviewer for pointing out this issue. We have normalized the histogram to the total number of photons. In this way, we generate a probability density that underpins the advantage of using the OFA. Figure 2g in the main manuscript has been changed accordingly.

Comment R2.4: Page 4, line 116. When authors compare the FCS results, they list four effects: “1) the number of photons per unit of time is higher, 2) the average brightness is larger ...” What is the explicit difference of 2) from 1)?

Answer R2.4: We agree with the referee that this distinction, which was meant to emphasize a visual impression, is confusing. We have now changed the manuscript to merge points 1) and 2):

- 1) the average brightness is larger (i.e., the number of photons per unit of time is higher), 2) the burst last longer, and 3) the overall number of recorded bursts is considerably larger than in the case of diffusion in an open solution. While the first effect stems from optical phenomena, the latter ...

Comment R2.5: Figure 3a. The horizontal and vertical axes are not marked.

Answer R2.5: We thank the reviewer for pointing this out. Figure 3a has been updated and the previously missing axes now read y : “Free energy G ” , and x : “Nuclear coordinate”

Comment R2.6: Figure 3. What is exactly the physical nature of the “dynamic fraction” depicted by violet? Resolving its population is the highlight of the sm-MFD study on Holliday junctions, yet the discussion of the “dynamic fraction” is a little vague. Suppose within this state the switch happens frequently as the name suggests, why is the corresponding cluster not further deviating from the static FRET line in Figure 3d compared to those of the HF and MF states? And why is the corresponding FRET efficiency spans much larger than those of the HF and MF states in Figure 3e,f?

Answer R2.6: To further explain our findings about the HJ FRET efficiency states, we have improved our Figs. 3c and 3d, and added more information about the dynamical behavior of the HJ in the main text:

We also display two kinds of FRET lines as graphical guidance [33, 34]. Molecules with no conformational exchange during the burst duration lie on the static FRET line (black) while molecules that exhibit conformational exchange (dynamic molecules) are shifted to the dynamic FRET line (cyan).

A bit later we have also added:

... that exhibit a shift to the dynamic FRET line. It is expected that molecules exhibit conformational exchange during their dwell time because the burst durations with OFA are longer than the average lifetime of one conformer. These results verify that extended observation times (i.e. improved recurrence analysis of single particles (RASP) analysis [21]) and larger brightness (i.e. lower shot noise) in the antenna provide insight into slower dynamical processes than those accessible in bulk studies (see Supplementary Fig. 17).

Moreover, the following text has been added to the caption of figure 3 of the main text:

The dynamic FRET line between the two FRET species is shown in cyan.

Comment R2.7: PART 1: Page 5, line 133. The authors performed Monte Carlo simulations and found “the probability to return to the observation volume is approximately 9 times larger in the quasi-2D geometry of an OFA with a channel height of 500 nm than for the case of an open solution”. The illustration from SI section C seems to imply that the recurrence is largely due to the spatial confinement in the z direction since the distance between the micropipette wall and the cover glass is only 100 nm. It seems that, for relatively large biomolecules, the signals in FCS experiments could mostly come from the same molecules within a very small OFA solution volume. Therefore, the emitters in OFA may not be in free diffusion like those in the open solution case. It would be good if the authors can comment on this and the effect of the spatial confinement of OFA on sampling of single emitters in solution.

Answer R2.7 PART 1: We thank the reviewer for the thoughtful and valuable points raised here. These features of the OFA are very interesting and potentially very helpful in various studies. Indeed, the recurrence observed in the simulations is due to the confinement in the z-direction, and the simulations indicate that the very same molecule is measured more often when using an OFA as compared to a bulk-like scenario. Other experimental approaches have also exploited the recurrence effect for analyzing single particles in solution. We mention the recurrence effect: Ref. 21 (Hoffmann, A. et al. Quantifying heterogeneity and conformational dynamics from single-molecule FRET of diffusing molecules: recurrence analysis of single particles (RASP), *Phys. Chem. Chem. Phys.* 13, 1857–1871 (2011)). We added exemplary RASP-data in Supplementary Fig. 17 in Supplementary Note 13. We can demonstrate that measurements with OFA have more recurrence events and lower shot-noise, so that the time window for the analysis of very slow exchange kinetics is clearly expanded.

Supplementary Fig. 17. **Recurrence E contour plots of the HJ at 1 mM MgCl₂ for measurements without and with OFA, respectively.** The FRET efficiency (E) trajectories within a burst were binned in time windows of 0.3 ms. The FRET efficiency E of the start bin is compared to the E-value of a second time window that is shifted by the given number of time bins of 0.3 ms. For the shift by one time bin of 0.3 ms the number of recurrence events (y-axis) without (a) and with OFA (b) does not differ significantly. However, due to the higher mean number of detected photons, N_F , with the OFA, the shot noise and the corresponding width of the FRET efficiency histograms are lower. Most events lie on the diagonal that indicate the transition to other FRET efficiencies has taken place. At larger time shifts of 32 bins (i.e. 6.6 ms), the difference between measurements without (c) and with (d) OFA is obvious. Comparing both conditions, the number of events with OFA dropped to 1/7 (c) in comparison to 1/3 (d) with OFA. Moreover, most events are off-diagonal events, which indicates that transitions to distinct FRET efficiency species took place. This demonstrates that measurements with OFA have more recurrence events and lower shot noise, so that the time window for the analysis of very slow exchange kinetics is expanded.

In addition, if we consider the imperfect photophysical properties of most fluorophores, this effect might be a disadvantage. For example, if the molecule under observation is photo-bleached, the refilling time (time for a molecule to enter the antenna volume and cross the observation/excitation volume) will be larger than in the case of diffusion in bulk. This effect can be avoided by controllably moving the pipette upwards (making the water layer of the antenna larger) and thus allowing new/fresh molecules to come in. To rule out potential dye blinking problems, we monitored the physical dye properties and evaluated the macroscopic photon arrival times after donor excitation (TGX) and after acceptor excitation (TRX) (see Figure R2). The method is described in: Kudryavtsev, V., Sikor, M., Kalinin, S., Mokranjac, D., Seidel, C. A. M., Lamb, D. C.; Combining MFD and PIE for accurate single-pair Förster Resonance Energy Transfer measurements. *ChemPhysChem* 13, 1060-1078 (2012). DOI:10.1002/cphc.201100822.

If there would be a signature of photobleaching, an asymmetry would be visible in the TGX-TRX distribution. We have set our experimental conditions (laser power, OFA geometry) such that we do not observe this. Based on Figure R2, we conclude a normal diffusional behavior of the sample and no significant acceptor blinking or photobleaching. Being aware that the photon budget of organic chromophores is limited, the experiments with an OFA benefit from the enhancement of the detection efficiency by a factor slightly larger than 2.2.

Figure R2. Experimental distribution of the macroscopic photon arrival times after donor excitation (TGX) and after acceptor excitation (TRX) for the HJ experiments with OFA.

To clarify these advantages, we added the following sentences to the discussion:

Measurements with an OFA allow one to expand the time range for studying biomolecular dynamics beyond the limit imposed by the translational diffusion time in a laser focus. Figure 4d demonstrates this for a transition within 50 μs with SNR=1.5 while our analysis predicts SNR=1 for measurements with a temporal resolution of 20 μs (see Supplementary Fig. 14d). On the slow side, we have measured up to about 1 ms, but application of the recurrence analysis of single particles (RASP) harbors the potential to reach tens of milliseconds [21]. In Supplementary Fig. 17, we show recurrence FRET efficiency plots without and with OFA, respectively, which demonstrate that measurements with OFA have more recurrence events and lower shot noise, so that the time window for the analysis of very slow exchange kinetics is significantly expanded. In addition, more efficient collection of photons by about 2.2 folds in an OFA gives access to the optimal photon budget, which is fundamentally limited by photobleaching.

PART 2: A follow-up on the free diffusion problem. It seems the water meniscus has a large contact area with air, so evaporation is probably unavoidable. Then it is likely that the solution continuously flows into the OFA to maintain the shape of the water meniscus, which also suggests that the molecules measured may not be in free diffusion. In SI section A the authors briefly mentioned controlling the pressure inside the micropipette by tuning the syringe plunger. It would be better if the authors can give more details, e.g., what pressure is desired to maintain the current shape of the water meniscus? Also, how will the pressure applied affect water evaporation within the OFA?

Answer R2.7 PART 2: Regarding water evaporation, our experimental observations strongly suggest that water evaporation can be neglected. First, we observe no change in the shape of the water meniscus over several hours while monitoring the interference fringes formed in the thin water layer (see new section of Supplementary Note 1 for more details). Second, we believe that the air inside the thin capillary will quickly saturate since it is a closed system. We have added the following passage to Supplementary Note 1 to address this issue.

We note that we do not observe any change in the shape of the water meniscus within the duration of the measurements, which typically take several hours (as long as the macroscopic drop of solution next to the micropipette is not evaporated). Therefore, we can safely assume no significant evaporation of the solution via the capillary. This process is also unlikely to influence the meniscus shape since the volume of air in the thin and long capillary is small and, thus, quickly saturates above the liquid surface.

Comment R2.8: Page 6, line 179. The authors claimed, “the antenna enhances the average photon detection rate by a factor of 4 for both donor and acceptor dyes while their ratio remains unchanged”. How was the quantitatively conclusion that the ratio remains unchanged from Figure 3b made? By comparing the linear regression fittings of the two DA clusters?

Answer R2.8: The FRET efficiency is defined as the ratio of the photons emitted by the acceptor to the sum of the photons emitted by the donor and the acceptor. In Fig. 3b of the main manuscript, we monitored the count rate of the donor and acceptor dyes and saw an equal enhancement for both. However, the FRET efficiency, which is displayed in Fig. 3c,d, did not change. Furthermore, in Supplementary Fig. 11, we directly compared FRET efficiencies obtained with and without OFA and could not see significant changes in the FRET efficiencies. We clarified this issue in the text by referring to the corresponding figure panels:

We verified that the antenna enhances the average photon detection rate by a factor of 4 for both donor and acceptor dyes (Fig. 3b) while the ratio FRET efficiency remains unchanged (Fig. 3c,d).

Comment R2.9: SI section M (now Supplementary Note 13, Figures 15a, e). Comparing Figure 13a,e it seems that when decreasing the Mg²⁺ concentration from 0.5 mM to 0.2 mM, the HF state population increases substantially. It is well-known that Mg²⁺ ions slow down the Holliday junction transitions, but why do they also alter the FRET state populations here? Do authors have an explanation on the data shown in Figure 13?

Answer R2.9: Upon increasing Mg^{2+} concentration, the level duration of the state MF increases more strongly than that of HF (32% compared to 11%, respectively), as shown in Figs. 13d,h and 13c,g. This indicates that MF is better stabilized by Mg^{2+} than HF. It is well-known that the equilibrium of Holliday junction is influenced by Mg^{2+} ions. This is also observed in the study reported in Ref: Clegg R. M., et al. (1992), Fluorescence resonance energy transfer analysis of the structure of the four-way DNA junction, *Biochemistry* 31:4846-4856. We also added more information about the dynamics to the main text, please see answer R2.6.

Reviewer #3 (Remarks to the Author):

This manuscript describes an interesting technical advance in single emitter/molecule measurements using an optofluidic antenna. Single molecule fluorescence measurements are powerful tools to extract distributions and dynamics in complex systems in biology and other scientific areas. While well established in many forms, the methods still have limitations. Along these lines, the authors devise an optofluidic antenna (combined with smMFD/PIE) that can result in multiple improvements to such measurements, including enhancements per passage in fluorescence signal (by improvements in collection efficiency and trajectory positioning in detection region) and diffusion time, as well as number of passages per molecule. The improvements are nicely demonstrated using the dynamics of a well-studied model system, the DNA 4-way junction. The work is carefully performed and described. The new method and capabilities will be interesting and useful for the scientific community. However, I have a few issues that should be addressed.

Answer R3: We are grateful to the reviewer for their thoughtful questions and constructive suggestions. We particularly appreciate their endorsement of the added value of our studies for single-molecule fluorescence analysis. Below we address all issues raised by the referee.

Comment R3.1: From the point of view of application of the method, one potential issue is the presence of large interfacial areas in the detection region. As is well established in the field (and noted in the manuscript), proteins and other biopolymers can have interactions with such interfaces, which can result in substantial perturbations to the properties of the system. While addressed somewhat, I think it is important for the authors to address this issue in more detail, and possibly suggest directions that may be used to alleviate it in future studies of more surface-sensitive systems.

Answer R3.1: As the referee points out and as we emphasize in our presentation, it is highly desirable to avoid surfaces when studying protein structure and dynamics. It is true that as compared to standard FCS measurements, our experimental arrangement provides more room for contact with surfaces. However, we point out that the data recorded in our experiments stem from the region that is fairly far from the glass substrate (at least tens of nanometer). Nevertheless, to minimize the effect of such interactions, we passivate all glass surfaces by coating them with BSA, following the procedure described in the literature (Hua, B. et al. An improved surface passivation method for single-molecule studies. *Nature Methods* 11, 1233–1236397 (2014); Paul, T., Ha, T. & Myong, S. Regeneration of PEG slide for multiple rounds of single-molecule measurements. *Biophys. J.* 120, 1788–1799 (2021)). We have added this information to the description of our method:

We passivated the cover glass surface by applying a 10 μ M concentrated BSA solution, following published protocols [48,49].

The upper OFA interface between the buffer and air does not involve a hard solid surface. We, thus, do not expect it to severely affect bonds and modify the structure of the biomolecules. Nevertheless, we acknowledge that this phenomenon is interesting in its own right, and as we point out in the discussion, OFAs open doors to future studies of the effect of air interfaces.

Comment R3.2: The interpretation of the purple populations in Figure 3 in terms of the structural model were unclear - authors please explain further. Also, why is this population higher for the OFA vs open geometries - is it intrinsic or related to surface effects?

Answer R3.2: We thank the reviewer for pointing this out. We have changed the main manuscript, please refer to answer R2.6.

Comment R3.3: It is possible that the 4wj accesses a distribution of states in transitions between the stable ones. How would this be handled in the analysis of higher time-resolution traces and can any insight be gained for such a situation?

Answer R3.3: We have added an analysis of our resolution limit in detecting two different FRET efficiency states in the main text and in the Supplementary Information (see Answer R1.1 to Referee 1). We can conclude that possible transition states in between the stable ones have to have a lifetime of much less than 100 μ s.

Comment R3.4: Since this is a major consideration for implementation of the OFA, it would be useful to include a little more detail of how the micropipette tip is positioned/held at the sub-micron distances from the coverglass surface (specifically how the absolute distance is measured/maintained).

Answer R3.4: We have extended Supplementary Information Note 1 to elaborate on more technical details, including tip positioning. Please also see Answer R1.2 to the comments of Referee 1.

Comment R3.5: In the section on 4wj slow dynamics, the authors note "These results verify that extended observation times and larger brightness in the antenna provide insight into slower dynamical processes than those accessible in bulk studies". This statement is somewhat unclear - what additional information about dynamics was obtained here with the OFA?

Answer R3.5: The measurements did not provide any additional information, but we presented the following benefits: we can (1) improve the precision of the data analysis (more photons, i.e., better signal-to-noise ratio (see R1.1)), (2) improve/shift the upper limit for the lifetime of an intermediate state and (3) improve the recurrence analysis of single particles (RASP) for slow exchange (see comment R2.7: PART 1 and Supplementary Fig. 17). In other words, as compared to the conventional confocal studies in bulk liquid, we expand the dynamic range of studies (μ s - ms) in both directions. In the manuscript, we write: "... in the FRET trace, we conclude that the transition path time is shorter than 50 μ s and that the lifetime

of a potential intermediate state must be smaller than this value". On the slow side, RASP harbors the potential to reach tens of milliseconds [21].

Moreover, we clarified the sentence as follows:

These results verify that extended observation times (i.e. improved recurrence analysis of single particles (RASP) [21]) and larger brightness in the antenna provide insight into slower dynamical processes than those accessible in bulk studies (see Supplementary Fig. 17).

Minor:

Comment R3.6: 50 microsecond binning times have been achieved previously in smFRET measurements using other techniques, e.g., doi.org/10.1021/ja9027023 or doi.org/10.1038/nmeth.1553. The authors may choose to include a couple of additional references in this regard.

Answer R3.6: We thank the referee for pointing out this issue. In fact, as reported in the Methods section, we have also used photostabilizers following the protocol in Campos et al, Nature Methods (2011). This is a helpful measure for addressing the inherent photophysical limitations of fluorophores such as photoblinking, which leads to early saturation of the signal, generating a signal bottleneck. We now explicitly refer to this in the manuscript by writing in the section on Fast HJ dynamics:

To suppress the inherent effects of photobleaching and blinking in fluorophores, we also added photostabilizers following the protocol of Ref.[36] (see "Experimental conditions" in Methods).

Furthermore, we comment:

We note that high temporal resolutions have also been reported in conventional arrangements solely through the use of photo-stabilizers and saturation of fluorophores at high excitation powers [36, 39, 40], but an OFA offers a decisive advantage because it provides higher count rates at lower excitation powers and, thus, less photodamage.

In these revised passages, we have included the references suggested by the referee.

We agree with the reviewer that high temporal resolutions have also been reported in conventional arrangements at high excitation powers through the use of photo-stabilizers. We point out, however, that the minimum useful binning time depends on the detected photon rate, i.e., on the experimental setup as well as the photophysics of the system under study. In our studies, we raised the laser power by approximately 13 times for an increased sampling rate, reaching a moderate irradiance of approximately 75 kW/cm². Under these conditions, signal saturation is still small and a quantitative analysis of all bursts is possible. Indeed, we have previously shown that under high irradiances photoionisation and triplet saturation of the dyes limit the fluorescence signal (Eggeling, et al., Anal. Chem. **70**, 2651 (1998); DOI: 10.1021/ac980027p). Moreover, we define a quantitative criterion for time resolution (eq. 12 and 13) via the signal-to-noise ratio (SNR). In this respect, our reported limited time resolution of 20 μs for SNR = 1 represents a very good value for our reported conditions.

In summary, our measurements demonstrate that OFA provides a decisive improvement through its 2.2 fold higher detection efficiency, which allows for quantitative single-molecule studies with high time resolution at moderate irradiances.

Comment R3.7: Is it feasible to perform RASP-type analysis for the slow 4wj experiments?

Answer R3.7: We thank the reviewer for mentioning this type of experiment. RASP-type analysis should be feasible for the experiments with and without OFA but this was not the scope of this paper. We have now added this nice possibility to the outlook section for further expanding the accessible time range (see also Comment R2.7, PART 1):

On the slow side, we have measured up to about 1 ms, but application of the recurrence analysis of single particles (RASP) harbors the potential to reach tens of milliseconds [21].

Comment R3.8: There appear to be missing error estimates for some of the values presented. Authors, please check and add as needed.

Answer R3.8: Error bars corresponding to the standard deviation of the signal have been added to Fig. 2b.

REVIEWERS' COMMENTS:

Reviewer #1 (Remarks to the Author):

I would like to thank the authors for the careful and detailed changes they have implemented to the manuscript. The authors have extensively addressed my concerns and I am (very) pleased with its current state. I strongly recommend publication of this manuscript for the significant advance it presents.

Reviewer #2 (Remarks to the Author):

The authors have adequately addressed my concerns. I am happy with this version of the manuscript.

Reviewer #3 (Remarks to the Author):

The authors have done a good job of addressing my previous set of comments. The developed method with improved capabilities will be of substantial interest and utility to the scientific community.